# A Receptor Story: Insulin Resistance Pathophysiology and Physiologic Insulin Resensitization’s Role as a Treatment Modality

**DOI:** 10.3390/ijms241310927

**Published:** 2023-06-30

**Authors:** Stanley T. Lewis, Frank Greenway, Tori R. Tucker, Michael Alexander, Levonika K. Jackson, Scott A. Hepford, Brian Loveridge, Jonathan R. T. Lakey

**Affiliations:** 1Eselle Health, La Jolla, CA 92037, USA; stanleytlewis2@gmail.com; 2Clinical Trials Unit, Pennington Biomedical Research Center, Louisiana State University, Baton Rouge, LA 77808, USA; frank.greenway@pbrc.edu; 3Department of Developmental and Cell Biology, University of California Irvine, Irvine, CA 92617, USA; trtucker@uci.edu; 4Department of Surgery, University of California Irvine, Orange, CA 92686, USA; michalex858@gmail.com; 5Well Cell Global, Medical and Scientific Advisory Board, Houston, TX 77079, USA; levy@wellcellsupport.com (L.K.J.); scott@wellcellglobal.com (S.A.H.); brian@diabetesrelief.com (B.L.); 6Department of Biomedical Engineering, University of California Irvine, Irvine, CA 92868, USA

**Keywords:** T2D, diabetes, carbohydrate metabolism, insulin resistance, physiologic insulin resensitization (PIR)

## Abstract

Physiologic insulin secretion consists of an oscillating pattern of secretion followed by distinct trough periods that stimulate ligand and receptor activation. Apart from the large postprandial bolus release of insulin, β cells also secrete small amounts of insulin every 4–8 min independent of a meal. Insulin resistance is associated with a disruption in the normal cyclical pattern of insulin secretion. In the case of type-2 diabetes, β-cell mass is reduced due to apoptosis and β cells secrete insulin asynchronously. When ligand/receptors are constantly exposed to insulin, a negative feedback loop down regulates insulin receptor availability to insulin, creating a relative hyperinsulinemia. The relative excess of insulin leads to insulin resistance (IR) due to decreased receptor availability. Over time, progressive insulin resistance compromises carbohydrate metabolism, and may progress to type-2 diabetes (T2D). In this review, we discuss insulin resistance pathophysiology and the use of dynamic exogenous insulin administration in a manner consistent with more normal insulin secretion periodicity to reverse insulin resistance. Administration of insulin in such a physiologic manner appears to improve insulin sensitivity, lower HgbA1c, and, in some instances, has been associated with the reversal of end-organ damage that leads to complications of diabetes. This review outlines the rationale for how the physiologic secretion of insulin orchestrates glucose metabolism, and how mimicking this secretion profile may serve to improve glycemic control, reduce cellular inflammation, and potentially improve outcomes in patients with diabetes.

## 1. Introduction 

Hormones that are released in an oscillating pattern (e.g., insulin) are physiologically designed to have receptors on the surface of effector cells that bind to their respective ligands. The cyclical release of these hormones plays a crucial role in maintaining homeostasis and regulating various physiological processes. In some cases, the ligand is brought into the cell where it separates from its receptor, and the receptor then migrates back to the cell’s surface, ready to bind with another ligand molecule. In the case of insulin, the receptor is activated as a tyrosine protein kinase, initiating a cascade of intracellular signaling events. The receptor is internalized as it undergoes an auto-phosphorylation sequence [1]. The reconfiguration of the receptor to its active extracellular configuration takes approximately 4 min after the ligand receptor complex formation, allowing for the subsequent binding and activation of additional ligand molecules [2]. This intricate process of ligand–receptor interaction, internalization, and reconfiguration plays a vital role in the precise control and regulation of insulin signaling.

For nearly 40 years, scientific research has established that pancreatic β cells secrete insulin in a dynamic biphasic manner by responding to a square wave increase in blood glucose concentrations. This periodic release pattern allows for optimal cellular carbohydrate metabolism, striking a delicate balance between providing sufficient insulin for glucose utilization, while avoiding overexposure of the insulin receptor and the consequent reverse feedback loop [3,4]. Intriguingly, insulin is released every 4–8 min or more commonly, 5–6 min, independent of the ingestion of food. The initial phase of insulin secretion involves the rapid release of insulin by β cells, which lasts for about 10 min. This rapid release of insulin by β cells is facilitated by the storage of insulin in miniature membrane-bound secretory granules that are primed to fuse with plasma [5]. The physiological release of insulin is vital for maintaining glucose homeostasis, and disruptions in this process during the initial phase have been linked to the development of type-2 diabetes (T2D) [6]. Following the initial phase, a second phase ensues, characterized by a plateau in insulin release that can last for 2–3 h. During the physiological release of insulin, the discrete oscillatory secretions and distinct trough periods stimulate ligand and receptor activation. The ability of insulin release to exhibit such oscillatory behavior can be attributed to the unique arrangement of β cells within the islets of Langerhans, which are cell clusters residing in the pancreas. Within these clusters, β cells maintain close contact with one another, forming intricate networks of autonomic nerves that enable the coordination of dynamic, cyclical patterns of insulin secretion [7]. Thus, the periodic pattern of insulin secretion is most likely a result of intrinsic β-cell mechanisms that become modified by exogenous stimulus such as hormones and neuronal inputs. 

## 2. Pancreatic Neuronal Network

Under normal physiological conditions, insulin oscillations are mediated by a complex pancreatic neuronal network that connects cells within the islets of Langerhans [8,9]. The brain and its neuronal network have a large influence on glucose homeostasis via the autonomic nerves that regulate the endocrine function of the pancreas. From tracing studies, neuronal networks have been mapped to pancreatic islets, which are innervated by neuronal circuits that come from the hypothalamus [10]. Within the pancreatic islets, both sympathetic and parasympathetic nerves of the autonomic nervous system can be found, indicating their involvement in modulating insulin release [11,12]. Furthermore, tracing studies also revealed that vagal afferent axons can be found within the pancreas [13]. Serotonin is a stimulating molecule for vagal afferent neurons and within the pancreas, β cells are known to produce serotonin to communicate with neighboring cells within the islet. It was recently thought that serotonin is a signaling molecule β cells use to communicate with the brain by vagal afferent neurons [10].

Within the neural network, an individual β cell is connected to other β cells, which mediates insulin secretion. β cells can be characterized as either “Leader cells” or “Follower cells” based on their functional characteristics [14]. Leader β cells possess pacemaker properties, allowing them to respond to glucose and synchronize communication patterns with follower cells. In response to glucose stimulation, leader β cells trigger a calcium (Ca^2+^) influx [15], which diffuses to the follower β cells. The influx of intracellular Ca^2+^ levels leads to β-cell depolarization, initiating electrical activity and a subsequent wave of action potentials. The precise coordination of β cell-to-β cell communication, facilitated by the interplay between Ca^2+^ influx and action potentials, is closely associated with the oscillatory pattern of insulin secretion [16].

## 3. Pancreatic Inflammation

An insult or injury (i.e., autoimmunity, obesity, toxin, trauma, stress etc.) causes inflammation in the pancreas that disrupts normal pancreatic insulin rhythmicity [17]. Type-one diabetes (T1D) is an autoimmune condition that results in islet inflammation caused by the infiltration of immune cells, leading to the progressive destruction of β cells. Individuals first diagnosed with T1D usually require small doses of insulin that eventually increases as the body begins to produce less insulin as β-cell mass decreases. Upon initial diagnosis of T1D, patients experience the honeymoon phase as β cells try to recover or compensate for the decline in β-cell mass [18]. The honeymoon phase is characterized as a partial remission that occurs shortly after a T1D patient begins insulin therapy. During this period, the individual’s diabetes may improve, and the patient may require less insulin than the first few days after diagnosis. The honeymoon phase can last approximately 3–12 months. Eventually, T1D patients will no longer experience normal physiological insulin secretion and will instead require lifelong exogenous insulin therapy via insulin injections or a continuous subcutaneous insulin pump. 

In contrast, T2D continues to produce insulin, but in an uncoordinated pattern without a true peak and trough waveform due to the inflammation and breakdown in the pancreatic neuronal network. Lang et al. found that healthy humans have intermittent insulin secretions and trough periods where T2D subjects had lost this native physiologic architecture [4]. It is thought that the disruption of insulin secretion may be a result of inflammation in the pancreas due to conditions such as obesity, toxins, trauma, etc. In early studies, it was found that insulin oscillates for approximately 15 min in fasting humans that do not have diabetes [19,20,21]. With an improvement in technology to measure insulin pulses, more recent studies have determined that the in vivo oscillation of insulin during fasting is closer to 5 min. For those who have T2D, Lang et al. revealed that insulin oscillations were shorter and highly irregular with a mean oscillation period of 8.8 min compared to control subjects who had a period of 10.7 min. Hunter et al. showed that control subjects had a higher glucose clearance and insulin sensitivity than those with T2D [22]. T2D subjects had greater insulin resistance and hepatic glucose output. In another study by Peiris et al., the number of insulin pulsations observed were correlated with a decrease in glucose clearance in diabetic subjects. These results supported the idea that abnormal insulin secretion and impaired insulin action are linked. This idea was driven by the hypothesis that a decline in insulin sensitivity is driven by insulin resistance (IR) [23]. 

## 4. Impaired Physiologic Pattern of Insulin 

As β cells cease to secrete insulin in the periodic synchronous pattern, insulin receptors undergo downregulation or internalization when they are continuously exposed to insulin. This continuous exposure disrupts the normal dynamics of insulin signaling and leads to a decrease in receptor responsiveness. A study demonstrated that subjecting healthy individuals to 20 h of constant insulin exposure at a steady glucose level resulted in a reduction in insulin action [24]. This prolonged exposure to insulin led to a decrease in insulin sensitivity and compromised insulin response [25].

It is important to recognize that hyperinsulinemia, or elevated levels of insulin, is relative to the normal insulin levels observed during intervals between periodic secretions (i.e., troughs). During routine fasted assessments, the absolute insulin level may fall within the broad normal range, making it challenging to detect impaired insulin function. This is why insulin impairment can often go undetected for an extended period, as conventional measurements may not capture the underlying abnormalities in insulin signaling and receptor dynamics.

Understanding the intricate relationship between insulin secretion, receptor responsiveness, and the consequences of continuous insulin exposure is critical in elucidating the mechanisms underlying IR and related metabolic disorders.

## 5. Impacts of Hyperinsulinemia 

Hyperinsulinemia occurs when the fasting concentration of insulin in the blood remains higher than normal levels for prolonged periods. Various factors can contribute to hyperinsulinemia, including incretin hormones, obesity, diet, visceral fat accumulation, as well as genetic and environmental factors (Figure 1) [26]. In rare cases, hyperinsulinemia can be attributed to specific conditions such as insulinoma, a tumor in β cells that leads to excessive insulin production, or nesidioblastosis, characterized by an abnormal increase in β-cell mass within the pancreas. One of the diseases most often associated with hyperinsulinemia is T2D, caused by progressive IR. In some cases of T2D, the body tries to compensate for IR by increasing the amount of insulin secreted by β cells in order to decrease blood glucose and maintain homeostasis. However, hyperinsulinemia itself can further induce insulin resistance over time because as described previously, constant exposure to insulin triggers a negative feedback loop, which causes insulin receptors to become less responsive to insulin and downregulate [27]. One study showed that when cells in vitro were chronically exposed to insulin, they had diminished insulin receptor tyrosine and serine autophosphorylation [28]. When chronic insulin exposure is removed, normal insulin receptor function can be achieved only if exposed cells maintain a molecular memory prior to chronic insulin exposure. However, if cells are exposed to persistent chronic levels of insulin for long periods of time, the molecular memory of the cell becomes reprogrammed in such a way that causes IR. Thus, these cells do not recover their normal insulin response even after chronic insulin exposure is removed [29]. 

## 6. Inadequate Receptor Function 

Over time, inadequate insulin receptor function and insulin resistance can contribute to the development of various diseases, including diabetes and other metabolic disorders. It is well-established that unchecked IR is the underlying pathophysiologic precursor to the development of T2D. As described earlier, T2D can result from inadequate receptor function that is often linked to obesity, but there are genetic predispositions that can result in receptor dysfunction. However, it is worth noting that genetic factors can also play a role in receptor dysfunction, leading to impaired insulin signaling. Two notable examples of genetic involvement are the *IRS1* and *IRS2* insulin receptor substrate genes. These genes encode peptides that hold significant importance in insulin-signaling pathways. Studies have demonstrated that specific polymorphisms within these genes are associated with a decrease in insulin sensitivity, ultimately predisposing individuals to T2D [30,31]. These genetic variations can disrupt the normal functioning of insulin receptors, hindering their ability to effectively respond to insulin molecules. 

One metabolic disorder that can result from an increase in IR is lipotoxicity. Free fatty acids (FFA) are an increased risk factor for those who have IR [32]. FFA in the plasma is regulated by insulin. Under normal physiological conditions, FFA increases during fasting and decreases after meal consumption. In obese patients, FFAs are often high and can contribute to insulin resistance and T2D [33]. Additionally, lipodystrophy and adipose tissue dysfunction can also contribute to elevated FFAs in the plasma. The two FFAs that are primarily responsible for insulin resistance and a decrease in insulin sensitivity are diacyglycerol (DAG) and ceramide. However, physical exercise can reduce the amount of DAG and ceramide found in skeletal muscle, which can improve insulin sensitivity [34]. In addition, a recent meta-analysis showed that intermittent fasting may also improve insulin sensitivity [35]. Two causes for FFA elevation are often associated with hyperlipidemia and inflammation. It has also been shown that increased IR is associated with an increase in cholesterol synthesis and a decrease in the uptake of cholesterol [36]. When IR increases, there is a shift in cholesterol metabolism. Often, patients with IR have decreased high-density lipoprotein (HDL) cholesterol, increased low-density lipoprotein (LDL) cholesterol, and increased triglycerides levels [36,37,38]. Disruption to lipid metabolism has mostly been associated with T2D. However, lipid metabolism disorders have also been seen in those with T1D. Lipid disorders can result from poor glycemic control in those who have either T1D or T2D [39]. Disruptions to lipid metabolism have been associated with an increased risk of developing coronary artery disease (CAD) and various other cardiovascular diseases [40,41]. More broadly, insulin resistance has been associated with metabolic diseases that masquerade as neurologic disorders. Diseases such as Parkinson’s disease and Alzheimer’s disease are associated with insulin resistance (Figure 1) [42].

**Figure 1 ijms-24-10927-f001:**
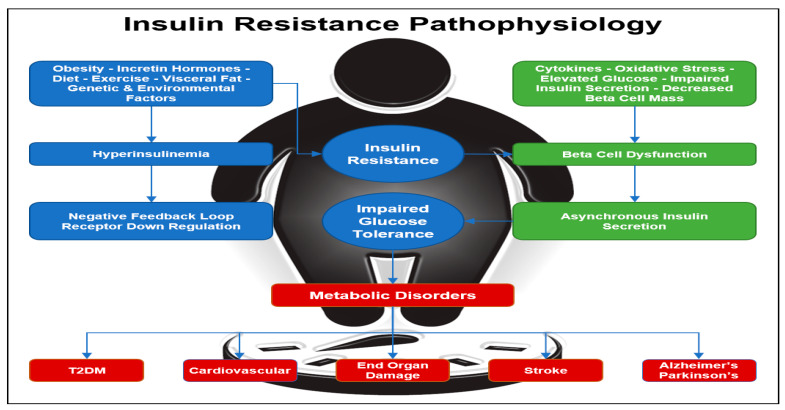
Insulin Resistance Pathophysiology. Insulin resistance can be initiated by obesity with an increase in visceral fat, inactivity, genetics, diet, and environmental factors. Insulin resistance leads to hyperinsulinemia with down regulation of the insulin receptors and a further increase in insulin resistance. Diabetes is a progressive disease with beta-cell dysfunction and loss of insulin pulses. Diabetes leads to diabetes complications, cardiovascular disease, metabolic syndrome, and is associated with neurodegenerative diseases [26,40,41,42].

## 7. Physiologic Insulin Administration 

One approach to counter IR would be to reintroduce precision physiologic insulin delivery. Currently, patients with T1D and T2D receive a continuous delivery of insulin either by insulin injections or a continuous subcutaneous insulin pump. IR often results from insulin receptors being constantly exposed to levels of insulin with no trough periods to allow the insulin receptor adequate time to reset. Counter to the goal of reducing serum glucose, the constant exposure to insulin can serve to create tolerance. Decreased sensitivity of receptors results in a decreased uptake of serum glucose and subsequent hyperglycemia. Often, type-2 diabetics are prescribed medications such as metformin or thiazolidinediones (i.e., rosiglitazone and pioglitazone) to combat abnormal insulin secretion caused by IR [43,44]. Although these medications are classified as insulin sensitizing, they do not restore or prevent abnormal insulin-secretion patterns. 

In 2012, Matveyenko et al. reported that insulin delivered in small intermittent boluses via the portal vein is more efficacious than receiving a constant insulin infusion [45]. In this study, rats received insulin continuously and in a more physiologic pattern of intermittent boluses. Rats that received a constant delivery of insulin had a delay and impairment in the activation of hepatic insulin receptors, which resulted in an impaired activation of downstream signaling. It is speculated that hepatic IR in diabetes is likely a result of the impaired physiologic insulin secretion experienced.

Physiological insulin resensitization (PIR) provides for insulin to be delivered peripherally in a dynamic pattern of physiologic boluses that mimics the healthy pancreas. This is achieved by inserting an intravenous access connected to a precision intravenous infusion pump that can be programmed to dynamically deliver insulin to mimic normal glucose metabolism. The physiologic administration of insulin consists of periodic cycling of up to 3 IU of regular fast-acting insulin infused intermittently between 4–8 min (usually 5–6 min), based on the body’s utilization for 2 to 4 h. These infusions are given based on the degree of insulin resistance, and precision dosing is undertaken to mimic hormonal signaling patterns in a concentration and frequency dependent on individual patient condition. During the infusion process, oral glucose is given to patients to simulate a meal and keep blood glucose levels in a prescribed range. Patients are observed during the process until glucose levels are stable after the physiologic insulin infusion is administered [46]. The rationale for the peripheral administration of IV insulin in a rhythmic pattern is to replace impaired physiological signals that are critical to cellular glucose metabolism. 

## 8. Insulin Receptor Upregulation 

Physiological insulin secretion exhibits a greater efficacy in upregulating insulin receptors compared to constant exposure to insulin. This effect is particularly notable in the liver, which experiences more insulin exposure than any other organ in the body due to the direct flow of insulin from the pancreas through the portal vein, which then spreads throughout the rest of the body [47]. Prolonged exposure to insulin leads to the downregulation of insulin receptors, ultimately contributing to IR [48]. The goal of PIR aims to address this issue by administering insulin in a manner that mimics the native pattern of hormone secretion. This is achieved by inducing metabolic activity to consume carbohydrates rather than fat. Glucose can be used and stored effectively when the body has not just enough insulin, but the right timing of insulin. However, when there is a lack of insulin or an over exposure of insulin over time causing a reverse feedback loop to downregulate the insulin receptor, a decline in carbohydrate metabolism occurs, and the body switches to fat metabolism in a process called ketosis. Ketosis triggers the production of ketones and, in severe cases, the accumulation of ketones can result in a life-threatening condition called diabetic ketoacidosis (DKA) [49]. Additionally, reducing excessive fat metabolism, through PIR, may counter the harmful effects of free radicals and oxidative stress that occur in the setting of elevated IL-6 levels found in T2D [50].

## 9. Cellular Glucose Uptake and Adenosine Triphosphate Production 

Restoring the physiologic cadence of insulin improves IR and allows for an increase in carbohydrate metabolism and a decrease in fat metabolism by restoring insulin’s ability to suppress lipolysis. As described previously, insulin binds to its receptor on cells, which are then endocytosed into the cell where glucose is further metabolized into “cellular energy”. Inside the cell, glucose undergoes a series of enzymatic reactions, primarily through the process of glycolysis. Initially, glucose is converted into pyruvate, which then enters the mitochondria to undergo additional processing into acetyl-CoA. This conversion occurs within the tricarboxylic acid (TCA) cycle, where acetyl-CoA participates in a series of chemical reactions. Along this cycle, the reducing agent NADH is produced, which is subsequently shuttled through complex I and II of the electron transport chain.

As electrons are transported, free energy is released, which actively pumps protons across the inner membrane of the mitochondria. As protons are pumped across the membrane, a proton gradient is formed, which creates an electrochemical gradient that fuels the production of adenosine triphosphate (ATP), commonly known as cellular energy (Figure 2) [51]. Thus, carbohydrate metabolism is essential for cells to produce cellular energy in the form of ATP.

## 10. Cellular Restoration 

Cell growth and proliferation are highly dependent upon carbohydrate metabolism. Glucose metabolism provides cellular energy (ATP), but it also supplies metabolites that are critical for the cellular synthesis of nucleic acids, proteins, and lipids [53,54]. One study showed that glucose metabolism is involved in the healing process of tendons [55]; a second study showed that glucose is important in human gingival fibroblast maintenance after periodontal surgery [56]. In the case of diabetes, blood glucose control and metabolism are important for wound healing. Furthermore, prolonged hyperglycemia in diabetes causes tissue to become energy-deficient, which over time can over result in a decline in wound healing [57]. For instance, the process of angiogenesis is required to promote tissue wound healing. Hyperglycemia decreases angiogenesis in endothelial cells, and it affects the stability of hypoxia-inducible factor 1-alpha (HIF-1a) to target genes such as the vascular endothelial growth factor (*VEGF*) to promote wound healing and tissue repair [58]. Often, diabetics with poor glucose control have wounds, most commonly foot ulcers, that go unnoticed, take longer to heal, become infected, or never adequately heal. Foot ulcers and their poor healing are often due to underlying neuropathy and a poorly functioning microcirculation. Nerve and microvascular tissue clearly benefit from optimized glucose ATP production and the related decrease in inflammation based on the body utilizing glucose as its primary fuel source instead of fat metabolism [50].

Glucose metabolism is essential for the development and maintenance of several tissues, including skeletal muscle, vasculature, liver, heart, and adipose tissue. The retina is one example of a tissue that is affected by the dysregulation of glucose metabolism. Diabetic retinopathy often results 10–20 years after the onset of diabetes due to glucose metabolism failure. Retinopathy is a disease of the microvascular system of the retina when the retina experiences hypoxia. Blood passes via the optic artery and perfuses the retina. In the retina, arteries radiate outward to form a capillary network that supplies blood vessels and oxygen. When high blood glucose levels persist, this can weaken and damage blood vessels within the retina. Damaged blood vessels starve the retina of oxygen, which leads to abnormal blood vessel outgrowth to compensate for oxygen deprivation. HIF-1a has a role in the stability of retina homeostasis, but intense fluctuations in carbohydrate metabolism can cause HIF-1a to fluctuate, which results in abnormal blood vessel outgrowth and progression of retinopathy [59]. 

Neuronal tissue appears to be the most sensitive tissue type affected when there is a decrease in energy from carbohydrate metabolism. Neuropathy in diabetes can result from glucose metabolism failure. Symptoms often include numbness, pain, muscle weakness, or tingling in the hands and feet. Hyperglycemia can induce mitochondria dysfunction/overload of glucose and increase reactive oxygen species (ROS) [59]. The accumulation of ROS results in induced cellular apoptosis and a decrease in nerve blood flow. Increased oxidative stress upregulates the polyADP-ribose polymerase (PARP) pathway, which increases inflammation and causes neuronal dysfunction. Neurovascular dysfunction from hyperglycemia affects five main pathways, including: the polyol pathway, the advanced glycation end-product (AGE) pathway, the protein kinase C pathway, the PARP pathway, and the hexosamine pathway [60]. One of the consequences of neuropathy is a decrease in wound detection and healing. To combat neuropathy, cell therapies have been used such as growth factors to promote neuron survival, function, and repair. Schratzberger et al. demonstrated that growth factors such as *VEGF*, *FGF2*, *NGF*, *BDNF*, and *IGF1* have neurotropic properties in treating diabetic neuropathy [61]. 

## 11. Discussion

Perhaps one of the most perplexing challenges in managing patients with diabetes and other metabolic disorders is IR. In patients with diabetes, the conventional approach to controlling serum blood glucose involves subcutaneous insulin administration or the use of oral medications that increase insulin production. However, this continuous exposure of insulin receptors to their ligand can lead to the development of tolerance. Continuous exposure to high levels of insulin causes insulin receptors to internalize, rendering them unavailable for engagement with insulin molecules. This internalization process contributes to the development of IR. Paradoxically, the administration of higher doses of insulin to reduce serum blood glucose levels can further worsen IR. This is due to a negative feedback loop, where higher levels of insulin in the bloodstream are associated with decreased availability of receptors, exacerbating the IR phenomenon. 

In the normal physiologic state, characterized by insulin sensitivity, insulin is secreted from pancreatic β cells in a specific cadence, with regular intervals of insulin secretion and troughs occurring approximately every 4–8 min. This intermittent exposure of peripheral insulin receptors to insulin plays a crucial role in maintaining the sensitivity of these receptors, preventing receptor tolerance and preserving insulin sensitivity. However, in patients who have developed diabetes or other IR phenotypes, the natural cadence of insulin secretion is disrupted. To address this, treatment strategies have been developed to mimic the physiologic intermittent secretion of insulin. These treatments, known as PIR, aim to restore the normal pattern of insulin exposure to peripheral insulin receptors. By administering insulin in a manner that resembles the natural insulin secretion pattern, PIR treatments have been associated with improvements in fasting serum glucose levels and HbA1c. 

Beyond these assessments that correspond to improvements in diabetes management, improvements in diabetic complications have also been observed. The rationale for improvements in conditions such as diabetic neuropathy, retinopathy, nephropathy, etc., may be related to improvements in glucose utilization and signaling associated with the physiologic administration of insulin. That is, insulin administration that mimics normal physiology appears to temporarily restore signaling that results in enhanced energy production from improvements in carbohydrate metabolism. While empiric assessments of insulin receptor availability and function are lacking, the apparent improvements in insulin sensitivity may be related to receptor function. The phenotype of IR appears to be a story of receptor availability mediated by the intermittent administration of insulin. Mimicry of this cadence of peaks and troughs appears to relieve IR and facilitate the improvement of complications associated with diabetes. 

## 12. Conclusions 

Insulin receptor modulation is a multifaceted process influenced and mediated, in part, by the intermittent peaks and troughs of the ligand. However, dysfunction in the autonomic nerves within the pancreas can disrupt the finely tuned oscillatory secretion of insulin, leading to abnormal insulin patterns and compromising glucose regulation. This disruption of the characteristic higher and lower concentrations observed in normal physiologic pancreatic β-cell secretion patterns is closely associated with insulin resistance, the onset of diabetes, and the subsequent development of complications and perhaps other disorders.

In contrast to conventional standard medicinal interventions, there is emerging evidence supporting the benefits of intravenous insulin administration that mimics the healthy pancreatic insulin secretion pattern. By restoring this physiological cadence, improvements in insulin resistance, restoration of metabolic function, enhanced cellular repair mechanisms, and a reduced risk of long-term complications have been observed. 

## Figures and Tables

**Figure 2 ijms-24-10927-f002:**
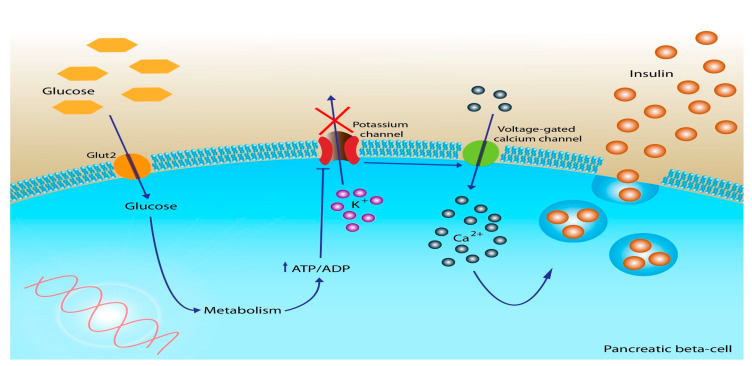
Cellular glucose uptake and ATP production [51,52]. Glucose is brought into the pancreatic beta cell by the GLUT-2 transporter and converted to glucose-6-phosphate by glucokinase initiating the glycolytic pathway and resulting in the generation of pyruvate. Pyruvate is transported into the mitochondria where ATP is generated through the electron transport chain. ATP depolarizes the ATP-sensitive potassium channel causing depolarization of the voltage dependent calcium channel, which causes prepackaged insulin granules to fuse with the cell membrane and release insulin into the blood stream (Image used under license from: https://support.shutterstock.com/s/article/Can-I-use-Images-on-my-website?language=en_US#:~:text=Shutterstock%20Photos%20Are%20Royalty%2DFree,multiple%20ways%20on%20multiple%20applications, accessed on 20 June 2023).

## Data Availability

Not applicable.

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
