# Peer review of "A Receptor Story: Insulin Resistance Pathophysiology and Physiologic Insulin Resensitization’s Role as a Treatment Modality"

_ijms, 2023, doi:10.3390/ijms241310927_

Round 1
Reviewer 1 Report
The review addresses a significant topic in the field of clinical medicine for the treatment of diabetes mellitus. However, I thought it needed to explore more about insulin resensitization. Therefore, I made considerations so that the authors can consider it and seek to improve the study.
1- Is T2D the consequence of IR or is IR the consequence of T2D?
2- The authors describe that physical exercise decreases DAG and ceramide levels in skeletal muscle, which improves insulin sensitivity. Regarding intermittent fasting what would be observed effects? Individuals adept t intermittent fasting, do they show improvement in insulin sensitivity?
3 – “Reducing fat metabolism through PIR offers additional benefits by mitigating harmful effects such as increased oxidative stress and the production of free radicals that can damage endothelial tissue”. How can PIR mitigate increased oxidative stress?
4 – Page 7: “Often, diabetics with poor glucose control have wounds, most commonly foot ulcers, that go unnoticed, take longer to heal, become infected, or never adequately heal. Foot ulcers and their poor healing are often due to underlying neuropathy and a poorly functioning microcirculation. Nerve and microvascular tissue benefit from optimized glucose ATP production and the related decrease in inflammation based on the body utilizing glucose as its primary fuel source instead of fat metabolism”. Please add a reference.
Reviewer 2 Report
This article has discussed the pathophysiology of insulin resistance and the potential improvement of patients treatment with diabetes. The authors have talked about the insulin oscillating release pattern, the way of insulin working with the receptors and the pancreatic neuronal network which control the release pattern. They also explained that the inflammation in the pancreas would disrupts the normal rhythm of insulin release which can cause diabetes. The impact of hyperinsulinemia and the abnormal receptor function also have been discussed in detail. The authors also explore the different approaches which could deal with insulin resistance and the recovery of receptor function and carbohydrate metabolism. The overall impression is that this is a great work for summarizing all the knowledge of Insulin Resistance, it explained the insulin working process and the cause of insulin resistance forming in detail. This article also compared different treatments and pointed the potentially improve outcomes in patients with diabetes. Only several minor suggestions: 1) Some acronyms (e.g. IR etc) wither occur prior to their full form or never mentioned (i.e. T2D) 2) There is no useful annotation for each figure. It would be better to add some sentences what information need to be emphasized in each figure. 3) Please check carefully for the proper citation, for example, please add proper citation for sentence: ‘Typically, for patients with T2D these infusion treatments are conducted on a weekly basis initially and then spaced out to one infusion between 4-6 weeks apart depending on patient response.’
